# Serum uric acid level and all-cause and cardiovascular mortality in peritoneal dialysis patients: A systematic review and dose-response meta-analysis of cohort studies

**Ting Kang[1☯], Youchun Hu[1☯], Xuemin Huang[1], Adwoa N. Amoah[1], Quanjun Lyu[1,2]***

**1** Department of Nutrition and Food Hygiene, College of Public Health, Zhengzhou University, Zhengzhou, Henan, China, **2** Department of Nutrition, the First Affiliated Hospital of Zhengzhou University, Zhengzhou, Henan, China

☯ These authors contributed equally to this work.
* lvquanjun666@163.com

## Abstract

### Background

The association between serum uric acid (SUA) and all-cause and cardiovascular disease (CVD) mortality in peritoneal dialysis (PD) patients is controversial. Therefore, we aimed to determine the relationship between SUA and all-cause and CVD mortality in PD patients.

### Method

Web of Science, EMBASE, PubMed and the Cochrane Library databases were searched from their inception to 7 April 2021. Effect estimates were presented as hazard ratios (HRs) with 95% confidence intervals (95% CIs) and pooled using random effects model.

### Result

Thirteen cohort studies with 22418 patients were included in this systematic review, of which 9 were included in the meta-analysis. Before switching the reference group, pooled result for the highest SUA category was significantly greater than the median for all-cause mortality (HR = 2.41, 95% CI: 1.37–4.26). After switching the reference group, the highest SUA category did not demonstrate an increased all-cause (HR = 1.40, 95% CI: 0.95–2.05) or CVD (HR = 1.30, 95% CI: 0.72–2.34) mortality compared with the lowest SUA category. Dose-response analysis suggested a nonlinear association between SUA and all-cause mortality risk ($P_{nonlinearity}$ = 0.002).

### Conclusion

This meta-analysis didn't find the relationship between SUA levels and all-cause and CVD mortality risk in PD patients. More rigorously designed studies are warranted in the future.

**Data Availability Statement:** All relevant data are within the paper and its Supporting information files.

**Funding:** The author(s) received no specific funding for this work.

**Competing interests:** The authors have declared that no competing interests exist.

## Introduction

Chronic kidney disease (CKD) is a worldwide public health problem with high incidence rate [1, 2] and high mortality [3], which have aggravated the burden of medical care. The consequences of CKD include cardiovascular disease, stroke, end-stage renal disease, renal replacement therapy (RRT) and kidney transplantation, all of which are serious and costly medical events [3]. The global prevalence of CKD is likely to rise further with the aging of the population and the increasing prevalence of diabetes, especially in China [4]. In the treatment of chronic kidney disease, dialysis is a conventional treatment method, including peritoneal dialysis (PD) and hemodialysis (HD). Compared with in-center HD, the use of PD is a more economical dialysis modality, which may potentially decrease infection risk, enhance patient satisfaction and preserve residual renal function, while having a comparable survival rate [5]. Due to the limited health-care resources, combined with the advantages of PD treatment and the support of government policies, PD has been widely used worldwide, including in China, Thailand and the United States [3, 6].

There are many factors that affect the survival rate of patients receiving dialysis treatment, including residual renal function, serum uric acid (SUA), and so on [7]. Uric acid (UA) is the final product in the liver from the degradation of dietary and endogenously synthesized purine or nucleotide compounds [8], about two-thirds of which is excreted by glomerular filtration [9]. Studies have found that UA is closely related to many chronic diseases. Epidemiological evidence demonstrated that higher UA concentration was a strong independent predictor of the incidence of type 2 diabetes mellitus (DM) [10, 11]. The possibility of gout arthritis development is correlated with the levels and the duration of SUA elevation [3]. Persistent hyperuricemia is closely related to cardiovascular, urolithiasis, thyroid dysfunction, psoriasis and hypertension [3]. In addition, research has shown that hyperuricemia is associated with all-cause and cardiovascular disease (CVD) mortality in CKD [12], HD [13] and PD patients [14, 15]. However, the evidence is conflicting.

Here we only focus on the relationship between SUA concentration and mortality (all-cause and CVD) in PD patients. Several studies showed that hyperuricemia was an independent risk factor for all-cause mortality in PD population [14–16], high level of SUA was associated with a high risk of CVD mortality in men treated with PD [15]. However, Lai et al. [17] found that there was an inverse association between the elevated SUA level and all-cause and CVD-associated mortality in women treated with continuous ambulatory PD. In addition, another study reported that hyperuricemia was weakly associated with all-cause and CVD mortality in PD patients [18]. Interestingly, different researchers have tried to explain the relationship between SUA levels and mortality from different directions. For example, different forms of SUA (time-averaged uric acid (TA-UA) [19] had been calculated at 3 months [16] or 6 months [20] after initiating PD; the longitudinal change in SUA [21]) were used to explore the association between mortality and SUA. Unfortunately, the impact of SUA on the survival of PD patients remains unclear.

To date, there has been no systematic review and dose-response meta-analysis to investigate the relationship between SUA and all-cause and CVD mortality in patients who had undergone PD. Hence, the objective of this study is to determine the association between SUA and all-cause and CVD mortality with a detailed analysis of eligible literature.

## Materials and methods

### Search strategy

One author (Kang) conducted systematic literature searches in electronic databases with no restriction on the language from their inception to 7 April 2021, including Web of Science,

EMBASE, PubMed and the Cochrane Library databases. An initial literature search in the mentioned databases used such keywords as "Peritoneal dialysis" and "Uric acid". The following search criteria were applied for PubMed: (("Peritoneal Dialysis"[Mesh]) OR (((Dialyses, Peritoneal) OR (Dialysis, Peritoneal)) OR (Peritoneal Dialyses))) AND (("Uric Acid"[Mesh]) OR ((((((((((((((Acid, Uric) OR (2,6,8-Trihydroxypurine)) OR (Trioxopurine)) OR (Potassium Urate)) OR (Urate, Potassium)) OR (Acid Urate, Ammonium)) OR (Urate, Ammonium Acid)) OR (Sodium Acid Urate)) OR (Urate, Monosodium)) OR (Urate, Sodium)) OR (Acid Urate, Sodium)) OR (Urate, Sodium Acid)) OR (Sodium Urate Monohydrate)) OR (Monohydrate, Sodium Urate)) OR (Urate Monohydrate))). Besides, the list of reference literature relevant was checked to identify additional eligible studies. All non-English studies were translated by software first, and then the researchers checked whether they could be included. Two reviewers (Kang and Hu) independently screened the abstracts to determine if they met the inclusion criteria, and disagreements were resolved through a third investigator (Huang). The searching strategies for the remaining databases are presented in the S1 Table.

## Inclusion criteria

The literature was selected if they met all of the following criteria: (1) the interest design were case-control or cohort studies; (2) study objects and interventions: participants treated with PD with no gender, race, or nationality limitations imposed; (3) the outcomes of interest were all-cause mortality and CVD mortality; (4) literature from which hazard ratios (HRs) data or calculate HRs data could be extracted were included in this meta-analysis. (5) If there were duplicate publications, the one with the largest number of participants or the longest follow-up period was included.

## Data extraction

The information was tabulated including the first author, year of publication, national/region where the research was conducted, study design, number of center, sample size, number of all-cause and cardiovascular deaths, follow-up duration, mean or median of SUA concentration, multi-factorial adjusted HRs and its 95% confidence intervals (CIs) of all-cause mortality or CVD mortality, adjusted covariates and quality evaluation information. Two authors (Kang and Hu) performed data extraction independently following the table contents. Discrepancies were resolved by discussion or a third investigator (Huang).

## Quality assessment of included studies

The quality of literature was evaluated independently by two authors (Kang and Huang) using the Newcastle-Ottawa Scale (NOS) [22], and the disputes were conformed through discussion with Hu. There are 3 quality parameters of NOS, of which the study population selects parameters worth up to 4 points, comparability parameters worth up to 2 points, and exposure or outcome evaluation parameters worth up to 3 points. The full score of NOS is 9, and studies with scores of 0 ~ 3, 4 ~ 6, and 7 ~ 9 points are defined as low, medium, and high quality ones, respectively.

## Data estimation

In this meta-analysis, SUA concentration was given in mg/dL. To use morbidity data to analyze possible dose-response relationships, it is necessary to have the following information for each SUA concentration category: assigned average or median SUA levels, deaths, follow-up person-years, adjusted HRs and 95% CIs. When the number of all-cause and CVD deaths in

each subgroup was not directly available in published data, appropriate statistical methods were employed to estimate missing data using the total number of deaths, HRs and the total number of patients in each subgroup [23]. For each SUA category in each study, the "years of follow-up" was calculated by multiplying the number of patients in that SUA category by the median or average follow-up months and dividing by 12. When the HRs and 95% CIs reported in the original study were not based on the lowest SUA group as the reference, we recalculated the relevant HRs and 95% CIs using the lowest dose group as the reference by the method developed by Hamling et al. [24].

After EXCEL conversion, the number of pseudo-effective cases, effect size and 95% CIs were used to replace the data provided by the original literature for dose-response meta-analysis [25]. If other relevant information was not available, we contacted the corresponding author via email.

## Statistical analysis

This meta-analysis was conducted using Stata software12.0 (version 12.0; Stata Corp, College Station, TX). We conducted a comparison between different SUA levels among included studies, including the highest SUA level compared with the lowest category of SUA level, and the highest / lowest category SUA level compared with the median SUA level. Given studies have reported several different possible relationships (U-shaped [19], inverse [17] or no relationship [18]) between SUA levels and all-cause or CVD mortality, we performed a dichotomy and dose-response meta-analysis. Multi-factor adjusted HRs and 95% CIs were extracted from the included studies and the pooled HRs was calculated using the lowest dose group as reference using the Mantel-Haenszel method developed by Hamling et al. [24]. A two-stage fixed-effects dose-response model was employed to explore the dose-response relationship between SUA levels and mortality in patients treated with PD. The potential linear or nonlinear relationship between SUA concentration and mortality (all-cause and CVD) was assessed using a restricted cubic spline regression model with 4 knots at fixed percentiles (5%, 35%, 65%, and 95%) of SUA concentration distribution [26]. The *P* value for curve linearity or nonlinearity was calculated by testing the null hypothesis that the coefficients of the second and third spline transformations were equal to zero. If $P < 0.05$, the null hypothesis was rejected and a nonlinear dose-response relationship was considered to exist. Otherwise, a linear regression model was considered.

The Q test and $I^2$ statistics were used for heterogeneity analysis. If the included literature were considered to have no significant heterogeneity ($P > 0.1$ and $I^2 < 50\%$), the fixed-effects model was applied. When heterogeneity was considered acceptable ($P < 0.1$ or $50\% \leq I^2 < 85\%$), the random effects model was used. When $I^2 > 85\%$, we considered that the results could not be pooled. Subgroup analysis was employed to explore the sources of heterogeneity. Sensitivity analysis, in which 1 study was removed at a time, was performed to evaluate the stability of the results. Egger's test was used to analyze the possibility of publication bias [27]. $P < 0.05$ was defined as significant publication bias. It must be mentioned that we did not register for this meta-analysis, but we conducted this systematic review and meta-analysis in strict accordance with the Preferred Reporting Items for Systematic Reviews and Meta-Analyses (PRISMA) statement.

## Results

### Literature search and selection

The PRISMA flowchart of the screening and selection process was summarized in Fig 1. A total of 971 references were identified to evaluate the relationship between SUA levels and

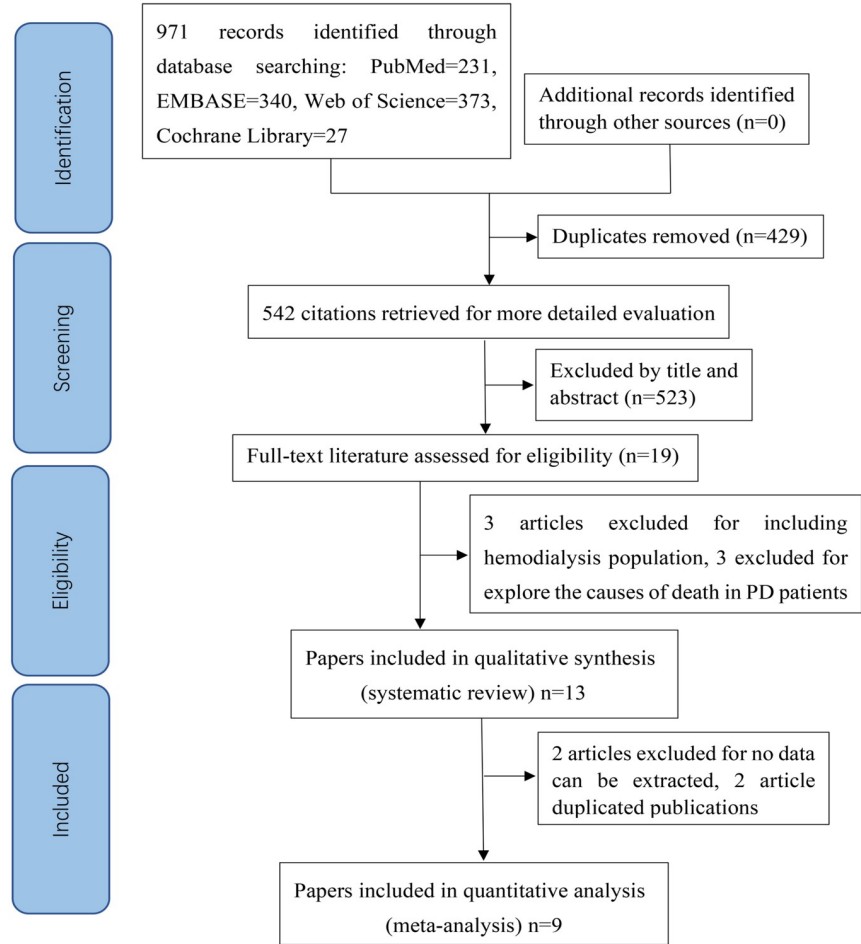

**Fig 1. Flow diagram of PRISMA presenting the process of search and selection of studies.** PRISMA, Preferred Reporting Items for Systematic Reviews and Meta-Analyses.

mortality during the initial search but 429 articles were removed due to duplication. A further 523 literatures were excluded for not meeting the inclusion criteria after screening the title and abstract. Nineteen articles were conducted full text assessment, and 6 studies were excluded, including 3 studies [28–30] excluded for the reasons of part of participants treated with hemo-dialysis and 3 studies [31–33] excluded for exploring the cause of mortality in patients treated with PD. Finally, 13 cohort studies with 22418 patients were included in this systematic review, of which 9 [14, 16–20, 34–36] were included in the meta-analysis. A study [21] was excluded in the meta-analysis because they focused on whether the longitudinal change in SUA affected all- cause mortality (SUA decliner vs SUA non-decliner) and another literature [37] explored the relationship between SUA and PD treatment failure. In addition, 2 studies [15, 38] dupli-cated publications with Xia [34].

## Study characteristics

The main characteristics of the included literature were presented in Table 1. A total of 13 studies consisting of 22418 participants were included. All studies were published from 2013 to 2021 and were designed as cohort studies, of which 9 were retrospective cohort studies (RCS) and three [14, 18, 35] were multicenter. Except for two studies [35, 36], all the studies

**Table 1. Characteristics and quality of included studies.**

| Study | Design | Region | Subjects | Number of center | Follow-up duration (months) | Outcomes | Concentration range of the SUA categories | Adjustment for covariates | NOS score |
|---|---|---|---|---|---|---|---|---|---|
| Feng, 2013 [16] | RCS | China | 156 | Single center | 31.3 | All-cause mortality | Group 1: ≤ 7.0 mg/dL | Age, HTN, DM, serum albumin, CRP, phosphate, RRF and UA group | 8 |
| | | | | | | | Group 2: 7.0–10.0 mg/dL (Reference) | | |
| | | | | | | | Group 3: ≥ 10.0 mg/dL | | |
| Dong, 2014 [18] | PCS | China | 2193 | Multi-center | 26.5 | All-cause, CVD mortality | Men: | Age, RRF, SA, hemoglobin, phosphate, CRP, CVD, BMI, mean arterial pressure, LDL-C and center size | 9 |
| | | | | | | | Tertile 1: 2.09–5.79 mg/dL (Reference) | | |
| | | | | | | | Tertile 2: 5.80–7.38 mg/dL | | |
| | | | | | | | Tertile 3: 7.39–16.7 mg/dL | | |
| | | | | | | | Women: | | |
| | | | | | | | Tertile 1: 1.74–5.37 mg/dL (Reference) | | |
| | | | | | | | Tertile 2: 5.38–6.65 mg/dL | | |
| | | | | | | | Tertile 3: 6.66–8.08 mg/dL | | |
| Xia, 2014 [15] | PCS | China | 985 | Single center | 25.3 | All-cause, CVD mortality | Men: | Age, BMI, Davies comorbidity score, hemoglobin, SA, SC, albumin-corrected calcium, SP, total triglyceride, LDL-C; RRF; log-transformed high-sensitivity CRP, total Kt/V, use of allopurinol, ACE inhibitor, or angiotensin receptor blocker and loop diuretics | 8 |
| | | | | | | | Tertile 1: ≤ 6.67 mg/dL (Reference) | | |
| | | | | | | | Tertile 2: 6.67–7.56 mg/dL | | |
| | | | | | | | Tertile 3: > 7.56 mg/dL | | |
| | | | | | | | Women: | | |
| | | | | | | | Tertile 1: ≤ 6.19 mg/dL (Reference) | | |
| | | | | | | | Tertile 2: 6.19–7.13 mg/dL | | |
| | | | | | | | Tertile 3: > 7.13 mg/dL | | |
| Xia, 2016 [34] | PCS | China | 1278 (diabetes:328 | Single center | 30.7 | All-cause, CVD mortality | Diabetic men: | Non-diabetes: age, BMI, history of hypertension and CVD, hemoglobin, SA, SP, SC, HDL-C, RRF, log-transformed high-sensitive CRP, use of allopurinol and Drugs used of allopurinol, ACE inhibitor, or angiotensin receptor blocker; Diabetes: non-diabetic adjustment content and glycated hemoglobin | 8 |
| | | | | | | | Tertile 1: < 6.46 mg/dL (Reference) | | |
| | | | | | | | Tertile 2: 6.46–7.38 mg/dL | | |
| | | | | | | | Tertile 3: ≥ 7.38 mg/dL | | |
| | | | | | | | Non-diabetic men: | | |
| | | | | | | | Tertile 1: < 7.00 mg/dL (Reference) | | |
| | | | | | | | Tertile 2: 7.70–7.89 mg/dL | | |
| | | | | | | | Tertile 3: ≥ 7.89 mg/dL | | |
| | | | | | | | Diabetic women: | | |
| | | | | | | | Tertile 1: < 5.89 mg/dL (Reference) | | |
| | | | | | | | Tertile 2: 5.89–7.09 mg/dL | | |
| | | | | | | | Tertile 3: ≥ 7.09 mg/dL | | |
| | | | non-diabetes:950 | | | | Non-diabetic women: | | |
| | | | | | | | Tertile 1: < 6.46 mg/dL (Reference) | | |
| | | | | | | | Tertile 2: 6.46–7.48 mg/dL | | |
| | | | | | | | Tertile 3: ≥ 7.48 mg/dL | | |

(*Continued*)

**Table 1.** (Continued)

| Study | Design | Region | Subjects | Number of center | Follow-up duration (months) | Outcomes | Concentration range of the SUA categories | Adjustment for covariates | NOS score |
|---|---|---|---|---|---|---|---|---|---|
| Hsieh, 2017 [37] | RCS | Taiwan, China | 371 | Single center | 36.7 | All-cause technique failure, peritonitis-related failure | Group 1: ≤ 8 mg/dL (Reference)<br>Group 2: > 8 mg/dL | Gender, age, BMI, comorbid conditions, and the use of ACE inhibitor, ARB, β-blocker, CCB, hypouricaemic agents, diuretics, BUN, creatinine, HB, ferritin, HbA1c, SA, Ca×P, GPT, RRF, icodextrin use, Balance dialysate use, assistance for dialysate exchanges, peritoneal Kt/V, weekly total Kt/V urea, nPNA, D/P (creatinine) at 4 hours, ultrafiltration, 24-hour urine output, and exit-site infection, tunnel infection, number of exchanges per day and peritonitis rate | 7 |
| Lai, 2018 [17] | RCS | Taiwan, China | 492 | Single center | 36.4 | All-cause, CVD mortality | Men:<br>Tertile 1: ≤ 6.8 mg/dL (Reference)<br>Tertile 2: 6.9–8.0 mg/dL<br>Tertile 3: ≥ 8.1 mg/dL<br>Women:<br>Tertile 1: ≤ 6.5 mg/dL (Reference)<br>Tertile 2: 6.6–7.6 mg/dL<br>Tertile 3: ≥ 7.7 mg/dL | Age, sex, BMI, the pre-dialysis status, smoking status, medications (ACE, ARB, ESA, furosemide, vitamin D, statin, allopurinol, CCB), comorbidities (DM, hypertension, CVD, Charlson score), PD related parameters (weekly total Kt/V urea, nPNA, D/P creatinine at 4 h, ultrafiltration, 24-h urine output, RRF), laboratory data (BUN, creatinine, albumin, GPT, WBC, alkaline phosphate, HB, ferritin, TSC, triglyceride, PTH, calcium, phosphate) | 8 |
| Zhang, 2018 [20] | RCS | China | 1063 | Single center | 33.0 | All-cause, CVD mortality | Group 1: < 7 mg/dL (Reference)<br>Group 2: ≥ 7 mg/dL | Age, Scr, P, Alb, BG, iPTH, history of DM, DBP, Charlson score | 8 |
| Chang, 2019 [19] | RCS | China | 300 | Single center | 22.6 | All-cause mortality | Group 1: TA-UA < 6 mg/dL<br>Group 2: TA-UA 6–8 mg/dL (Reference)<br>Group 3: TA-UA ≥ 8 mg/dL | Age, sex, DM, CVD, RRF, BMI, SBP, Hb, Alb, BUN, Cr, Na, K, $CO_2$, cCa, P, LDL-C, CRP, RASi, diuretic | 8 |
| Xiang, 2019 [14] | RCS | China | 9045 | Multi-center | 29.4 | All-cause, CVD mortality | Quintile 1: < 6.06 mg/dL<br>Quintile 2: 6.06–6.67 mg/dL<br>Quintile 3: 6.68–7.27 mg/dL (Reference)<br>Quintile 4: 7.28–8.03 mg/dL<br>Quintile 5: ≥ 8.04 mg/dL | Age, sex, BMI, DM, CVD, RRF, hemoglobin, SA, serum potassium, serum natrium, SP, serum calcium, serum parathyroid hormone, SC, and fasting plasma glucose | 7 |
| Chang, 2019 [21] | RCS | China | 309 | Single center | ≥4.0 | All-cause mortality | Group 1: SUA decliner<br>Group 2: SUA non-decliner | Gender, age, BMI, SBP, Hb, Na, K, Cl, BUN, Cr, $CO_2$, Ca, P, ALB, TG, FBG, CRP, RRF, PET type, Kt/V, CCB, RASi, β-blocker, diuretic | 7 |
| Xiao 2020 [38] | RCS | China | 802 | Single center | 68.7 | All-cause mortality | Group 1: > 7 mg/dL<br>Group 2: ≤ 7 mg/dL (Reference) | Age, gender, Charlson comorbidity score, PD vintage, total Kt/V, using of angiotensin-converting enzyme inhibitor or angiotensin II receptor blocker, using of diuretic, using of uric acid-lowering agent, total cholesterol, high-density lipoprotein cholesterol, neutrophil to lymphocyte ratio, intact parathyroid hormone, ECW/TBW ratio ≥0.4, ASMI groups, and ASMI groups × SUA, serum albumin | 8 |

(*Continued*)

**Table 1.** (Continued)

| Study | Design | Region | Subjects | Number of center | Follow-up duration (months) | Outcomes | Concentration range of the SUA categories | Adjustment for covariates | NOS score |
|---|---|---|---|---|---|---|---|---|---|
| Sugano 2020 [35] | PCS | Japan | 4742 | Multi-center | 12.0 | All-cause mortality | Group 1: < 5.0 mg/dL | Age, sex, dialysis duration, BMI, UV, use of ULT, diabetes, history of acute myocardial infarction, cerebral hemorrhage and cerebral infarction comorbid disease, and laboratory data including BUN, Cr, albumin, CRP, and Hb | 8 |
| | | | | | | | Group 2: 5.0 to < 5.5 mg/dL | | |
| | | | | | | | Group 3: 5.5 to < 6.0 mg/dL | | |
| | | | | | | | Group 4: 6.0 to < 6.5 mg/dL | | |
| | | | | | | | Group 5: 6.5 to < 7.0 mg/dL | | |
| | | | | | | | Group 6: 7.0 to < 7.5 mg/dL (Reference) | | |
| | | | | | | | Group 7: 7.5 to < 8.0 mg/dL | | |
| | | | | | | | Group 8: 8.0 to < 8.5 mg/dL | | |
| | | | | | | | Group 9: ≥ 8.5 mg/dL | | |
| Coelho 2020 [36] | RCS | Portugal | 682 | Single center | 31.4 | All-cause mortality | Not reported | Age, diabetes, comorbidity and baseline residual kidney function | 7 |

NOS, Newcastle-Ottawa Scale; RCS, retrospective cohort study; PCS, prospective cohort study; DM, diabetic mellitus; CVD, cardiovascular disease; RRF, residual renal function; BMI, body mass index; HTN, underlying hypertensive nephropathy; UA, uric acid; Alb, albumin; BUN, blood urea nitrogen; LDL-C, low-density lipoprotein cholesterol; HDL-C, high-density lipoprotein cholesterol; CRP, C-reactive protein; SA, serum albumin; SC, serum creatinine; SP, serum phosphorus; ACE, angiotensin-converting enzyme; ARB, inhibitors/angiotensin II receptor blocker; ESA, erythropoiesis stimulating agents; CCB, calcium channel blocker; nPNA, normalized protein nitrogen appearance; GPT, glutamic-pyruvic transaminase; WBC, white blood cell counts; PTH, intact parathyroid hormone; TSC, transferrin saturation, cholesterol, HB, hemoglobin; PD, peritoneal dialysis; ECW/TBW, extracellular water/total body water; ASMI, appendicular skeletal muscle mass index; SUA, serum uric acid; UV, urinary volume; D/P, dialysate-to-plasma; RASi, renin-angiotensinsystem inhibitor; Kt/V, urea clearance index; DBP, diastolic blood pressure; FBG, fasting blood glucose; SBP, systolic blood pressure; TG, triglyceride; PET, peritoneal equilibration test; ULT, urate-lowering treatment.

were conducted in China, two [17, 37] of which were carried out in Taiwan. The median or mean time of follow-up duration was 4.0 to 68.7 months. Five included studies explored the relationship between SUA and mortality by comparing the highest with the lowest SUA level, another four by comparing the lowest or highest with the intermediate level of SUA. The NOS score ranged from 7 to 9, and all included studies were considered to be of high quality (see S2 Table).

## Different SUA forms with all-cause and CVD mortality

UA values at different time points were used to explore the relationship with all-cause and CVD mortality. Eight studies [14–18, 34, 36, 38] used UA within 3 months of PD as a baseline to explore the relationship with mortality. Zhang et al. [20] compared the effect of high UA (≥ 420μmol/L) with normal UA (≤ 420μmol/L) on mortality after 6 months of PD. In addition, in a recently published study [19], the average time UA was calculated to investigate the association between TA-UA and all-cause mortality in PD patients, taking into account that UA concentration was easily affected by dialysis efficiency, diet, and medication. Furthermore, the effect of SUA decline and non-decline during PD on mortality has been studied by Chang et al. [21], showing that the decline of SUA meant a higher risk of all-cause mortality. In contrast, Feng et al. [16] found that unchanged SUA levels were associated with greater risk of death. A long-term observational cohort study [37] reported that the rate of all-cause technical failure was significantly higher in the hyperuricemia group than in the normal uricemia group.

Hyperuricemia was an independent risk factor for a higher risk of all-cause technical failure, which meant higher risk of death.

## Impact of UA on mortality in male and female treated with PD

The adverse effect of hyperuricemia on all-cause mortality was more prominent in the men's group, but there was no significant difference between male and female [14] on CVD mortality. For patients treated with continuous ambulatory PD, elevated UA levels were associated with reduced all-cause and CVD mortality in women, while there were no significant differences in men [17]. Hyperuricemia, however, is an independent predictor of all-cause and CVD mortality in males treated with PD [15].

## Impact of UA in PD population with diabetes or non-diabetes

The adverse effect of hyperuricemia on all-cause mortality was more prominent in patients without DM, while the effect was not significant between diabetes and non-diabetes subgroups for CVD mortality [14]. Another study showed that diabetes mellitus combined with peritoneal dialysis was an independent risk factor for death from cardiovascular events [20]. A multi-center cohort study of 2264 PD participants (37.7% of whom had DM) showed that each 1 mg/dL increase in SUA predicted a 10% increase in the CVD mortality rate in DM patients and a 12% increase in non-diabetic patients [18]. Elevated SUA was an independent risk factor for CVD mortality in males treated with PD, as well as predicted a higher risk of all-cause mortality in non-diabetic males, but not for females in predicting the risk of all-cause and CVD mortality [34].

## Relationship between SUA and all-cause mortality (before recalculated the HRs and 95% CIs)

**High vs low.**   Five studies [17, 18, 20, 34, 36] reported HRs and 95% CIs of all-cause mortality for the highest SUA category compared with the lowest. As presented in Fig 2a, all-cause mortality (HR = 1.19, 95% CI = 0.82–1.71, $I^2$ = 81.3%) was not significantly elevated compared with the lowest category of patients with PD. No obvious publication bias was found (t = -0.23, $P$ = 0.83).

**Sensitivity analysis.**   The combined HR changed (HR = 1.39, 95% CI: 1.08–1.79) and the heterogeneity was reduced after excluding Lai et al. [17] from the meta-analysis (from $I^2$ = 81.3% to $I^2$ = 56.9%), which could explain the part source of heterogeneity (see S1 Fig).

**High or low vs median.**   Four researches [14, 16, 19, 35] reported HRs and 95% CIs of all-cause mortality for the highest or lowest SUA category compared with the median. As shown in Fig 2b, the pooled result for the highest SUA category was significantly greater than the median (HR = 2.41, 95% CI: 1.37–4.26, $I^2$ = 72.0%). No significant publication bias was found (t = 3.62, $P$ = 0.07). But the lowest versus median levels of SUA were not associated with the all-cause death risk (HR = 1.45, 95% CI: 0.96–2.18, $I^2$ = 58.9%, Fig 2c). No significant publication bias was found (t = 0.98, $P$ = 0.43).

**Sensitivity analysis.**   The results of sensitivity analysis confirmed the stability of our result (see S2 and S3 Figs).

## Relationship between SUA and CVD mortality (before recalculated the HRs and 95% CIs)

**High vs low.**   Four studies [17, 18, 20, 34] reported HRs and 95% CIs of CVD mortality for the highest SUA category compared with the lowest. As shown in Fig 3, we did not find any

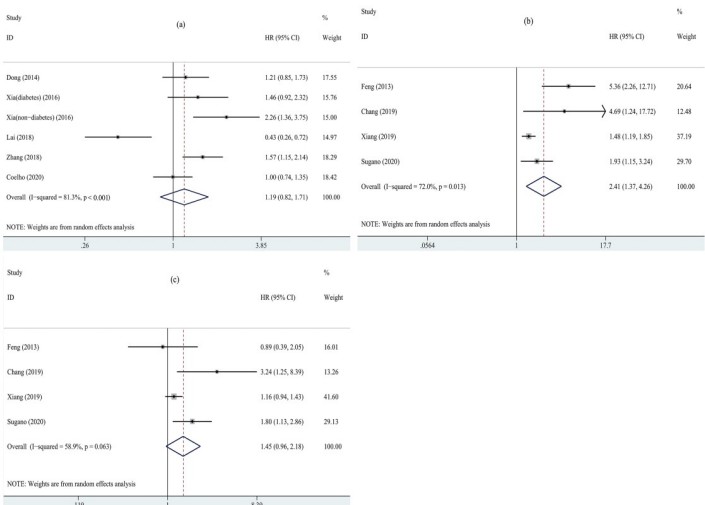

**Fig 2. Forest plots for relationship between SUA and all-cause mortality in PD patients.** (a) the highest SUA category versus the lowest. (b) the highest SUA category versus median. (c)the lowest SUA category versus median. SUA, serum uric acid; PD, peritoneal dialysis.

association of SUA level and CVD mortality (HR = 1.48, 95% CI: 0.80–2.74, I² = 79.6%). No obvious publication bias was found (t = -0.05, *P* = 0.96).

**Sensitivity analysis.** The combined HR changed (HR = 1.93, 95% CI: 1.39–2.68) and the heterogeneity was reduced after excluding Lai et al. [17] from the meta-analysis (from I² = 79.6% to I² = 13.9%), which could explain the part source of heterogeneity (see S4 Fig).

**High or low vs median.** Only one research [14] reported HRs and 95% CIs of CVD mortality for the highest (HR = 1.14, 95% CI: 0.79–1.67) or lowest (HR = 1.17, 95% CI: 0.82–1.66) SUA category compared with the median.

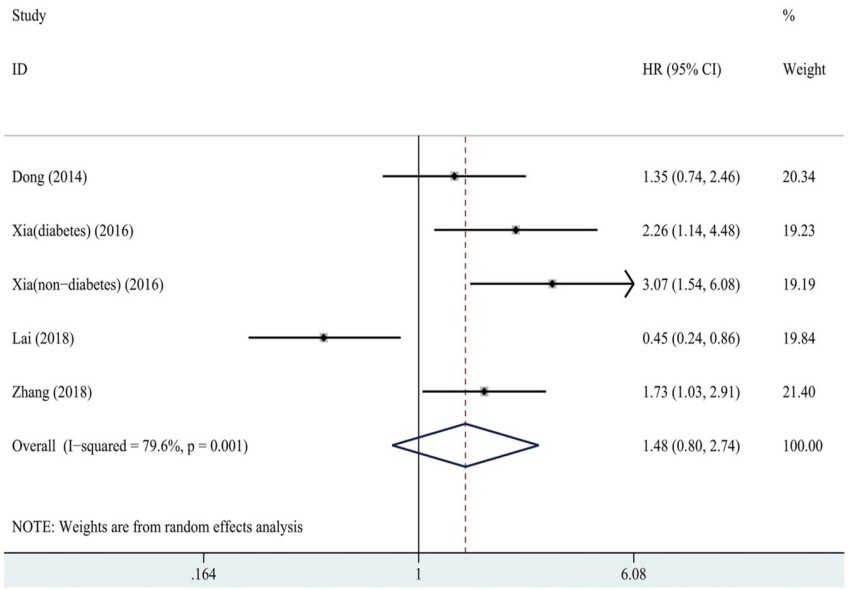

**Fig 3. Forest plot for relationship between SUA and cardiovascular mortality in PD patients.** The highest SUA category versus the lowest. SUA, serum uric acid; PD, peritoneal dialysis.

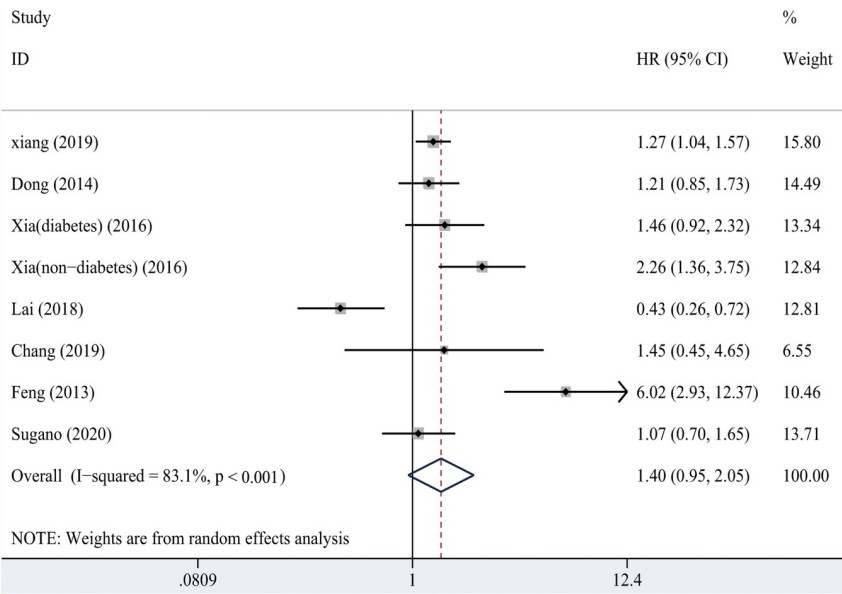

**Fig 4. Forest plot about the relationship between SUA and all-cause mortality in PD patients.** SUA, serum uric acid; PD, peritoneal dialysis.

## Relationship between SUA and all-cause mortality (after recalculated the HRs and 95% CIs)

The pooled HR and 95% CI of all-cause mortality comparing the highest versus the lowest category was 1.40 (95% CI: 0.95–2.05), presented in Fig 4. There was no significant difference in all-cause mortality between the highest and lowest subgroup of SUA level, which was consistent with the result before we recalculated. No obvious publication bias was found (t = 0.53, $P = 0.62$), but there was significant heterogeneity ($I^2 = 83.1\%$, $P < 0.001$) between the studies.

**Subgroup analysis.** Significant associations were found for subgroups by study design (PCS or RCS), number of center (multi-center or single center), publication years (2013–2016 or 2017–2020), sample size ($< 900$ or $> 900$), follow-up duration ($< 30$ months or $> 30$ months), male proportion ($< 50\%$ or $\geq 50\%$), or whether the results were adjusted for diabetes status and BMI (see Table 2).

**Sensitivity analysis.** The combined HR changed (HR = 1.62, 95% CI: 1.17–2.24) and the heterogeneity was reduced after excluding Lai et al. [17] from the meta-analysis (from $I^2 = 83.1\%$ to $I^2 = 73.0\%$), which could explain the part source of heterogeneity (see S5 Fig).

**Dose-response analysis.** The dose-response relationship between SUA and all-cause mortality was analyzed by random effect nonlinear model. Fig 5 reveals the nonlinear dose-response relationship between SUA concentration and all-cause mortality in PD population, suggesting higher SUA level was associated with increasing all-cause mortality ($P_{nonlinearity} = 0.002$).

## Relationship between SUA and CVD mortality (after recalculated the HRs and 95% CIs)

There was no significant difference in CVD mortality between the highest and lowest subgroups of SUA (HR = 1.30, 95% CI: 0.72–2.34) (Fig 6), which was consistent with the result (the high vs low) before we recalculated. No obvious publication bias was found (t = 0.72, $P = 0.52$), but there was significant heterogeneity ($I^2 = 80.8\%$, $P < 0.001$) between the studies.

**Table 2. Subgroup analysis of the relationship between serum uric acid and all-cause mortality.**

| | Serum uric acid | | |
| --- | --- | --- | --- |
| | **Number of study** | **HR (95% CI)** | **Heterogeneity (I²)** |
| **Study design** | | | |
| Prospective cohort study | 4 | 1.40(1.04, 1.88) | 46.0% |
| Retrospective cohort study | 4 | 1.44(0.57, 3.61) | 91.5% |
| **Number of center** | | | |
| Multi-center | 3 | 1.23(1.04, 1.45) | 0.0% |
| Single center | 5 | 1.63(0.70, 3.83) | 90.0% |
| **Publication years** | | | |
| 2013–2016 | 4 | 2.08(1.16, 3.73) | 82.4% |
| 2017–2020 | 4 | 0.93(0.55, 1.56) | 80.4% |
| **Sample size** | | | |
| < 900 | 4 | 1.50(0.50, 4.54) | 91.6% |
| > 900 | 4 | 1.33(1.05, 1.69) | 46.0% |
| **Follow-up duration(months)** | | | |
| < 30 | 4 | 1.23(1.05, 1.45) | 0.0% |
| > 30 | 4 | 1.67(0.63, 4.45) | 92.5% |
| **Male (%)** | | | |
| ≥ 50% | 6 | 1.75(1.17, 2.62) | 76.7% |
| < 50% | 2 | 0.73(0.27, 2.02) | 90.6% |
| **Adjust for diabetes** | | | |
| Yes | 5 | 1.33(0.69, 2.53) | 88.8% |
| No | 3 | 1.53(1.08, 2.18) | 48.9% |
| **Adjust for BMI** | | | |
| Yes | 7 | 1.18(0.86, 1.62) | 73.9% |
| No | 1 | 6.02(2.93, 12.37) | - |

HR, hazard ratio; CI, confidence interval; BMI, body mass index.

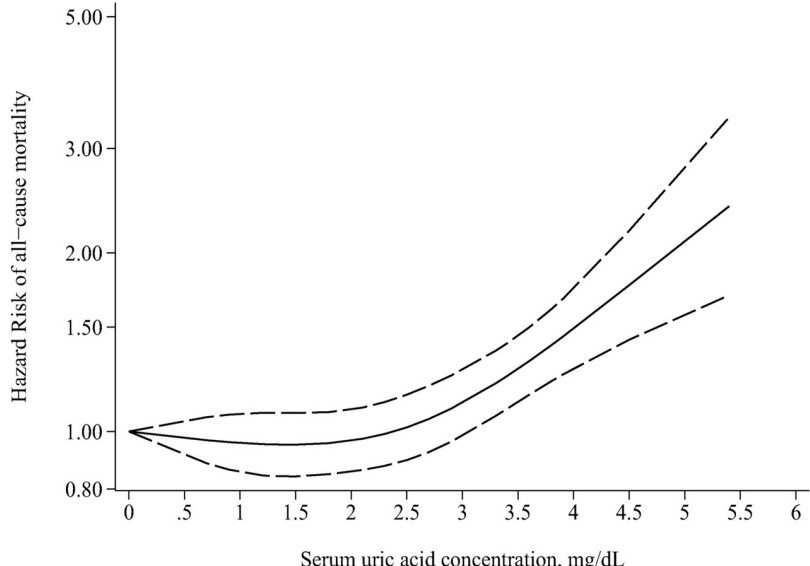

**Fig 5. Dose-response relation between SUA concentration and all-cause mortality in PD patients.** The solid line and the dash line represent the estimated hazard risk and its 95% confidence interval. SUA, serum uric acid; PD, peritoneal dialysis.

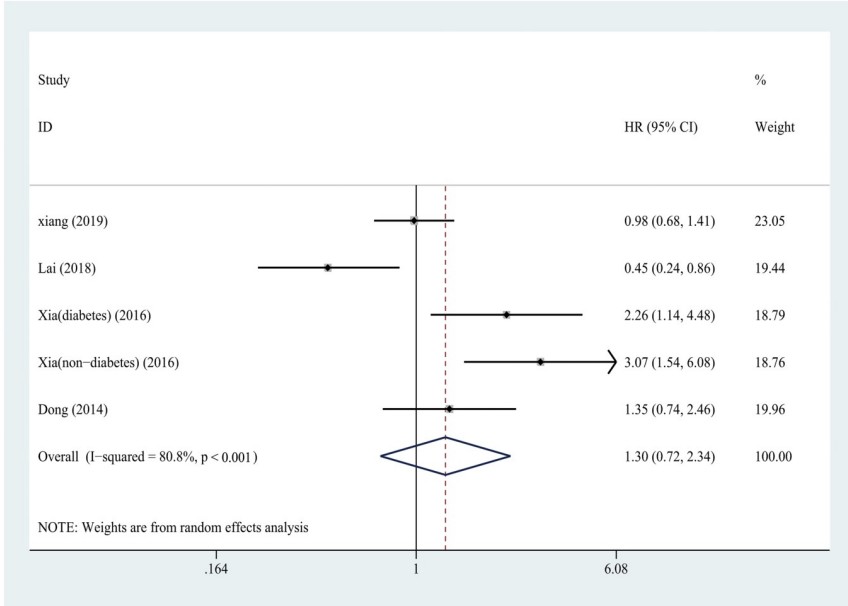

**Fig 6. Forest plot about the relationship between SUA and cardiovascular mortality in PD patients.** SUA, serum uric acid; PD, peritoneal dialysis.

**Subgroup analysis.** As shown in Table 3, significant associations were found for subgroups by study design, publication years, or whether the results were adjusted for diabetes status. Non-significant associations of SUA level with CVD mortality were detected in the subgroup analysis conducted by number of center, sample size, follow-up duration, and male proportion.

**Sensitivity analysis.** The results showed that the pooled HR value was close before and after removing any study, indicating that the result was relatively stable (see S6 Fig).

**Dose-response analysis.** The P values of the overall null hypothesis tests of model parameters were all $> 0.05$, and the P values of the goodness-of-fit tests of model were all $< 0.05$, indicating that neither liner ($P = 0.360$, random effect linear model) nor nonlinear ($P = 0.476$, random effect nonlinear model) relationship between SUA and CVD mortality was observed.

## Discussion

To the best of our knowledge, this is the first study to determine the association between SUA and all-cause and CVD mortality in PD patients based on dose-response meta-analysis of cohort studies. Before and after recalculating the relevant HRs and 95% CIs using the lowest dose group as the reference, the pooled results were consistent. We found that higher SUA level was significantly associated with higher all-cause mortality compared with the median level. In addition, the results from this meta-analysis did not detect any relationship between different SUA levels and the risk of all-cause and CVD mortality. The dose-response analysis suggested a J-shaped nonlinear relationship between SUA concentrations and all-cause mortality. When Lai et al.'s study was removed, we detected that compared with lowest SUA levels, highest SUA levels were associated with an increased risk of all-cause mortality in PD patients.

SUA was a clinically powerful nutritional marker and as well as an independent predictor of all-cause and CVD death risk [39]. A systematic review and meta-analysis of hemodialysis death risk factors published in 2017 showed that all-cause and cardiovascular death are

**Table 3. Subgroup analysis of the relationship between serum uric acid and cardiovascular mortality.**

| | Serum uric acid | | |
| --- | --- | --- | --- |
| | Number of study | HR (95% CI) | Heterogeneity ($I^2$) |
| **Study design** | | | |
| Prospective cohort study | 3 | 2.06(1.27, 3.34) | 38.7% |
| Retrospective cohort study | 2 | 0.70(0.33, 1.49) | 77.0% |
| **Number of center** | | | |
| Multi-center | 2 | 1.07(0.78, 1.45) | 0.0% |
| Single center | 3 | 1.45(0.44, 4.79) | 89.5% |
| **Publication years** | | | |
| 2013–2016 | 3 | 2.06(1.27, 3.34) | 38.7% |
| 2017–2020 | 2 | 0.70(0.33, 1.49) | 77.0% |
| **Sample size** | | | |
| < 900 | 2 | 1.00(0.21, 4.88) | 91.2% |
| > 900 | 3 | 1.52(0.80, 2.87) | 76.1% |
| **Follow-up duration(months)** | | | |
| < 30 | 2 | 1.07(0.78, 1.45) | 0.0% |
| > 30 | 3 | 1.45(0.44, 4.79) | 89.5% |
| **Male (%)** | | | |
| ≥ 50% | 3 | 1.81(0.85, 3.86) | 81.2% |
| < 50% | 2 | 0.78(0.27, 2.30) | 83.4% |
| **Adjust for diabetes** | | | |
| Yes | 2 | 0.70(0.33, 1.49) | 77.0% |
| No | 3 | 2.06(1.27, 3.34) | 38.7% |
| **Adjust for BMI** | | | |
| Yes | 5 | 1.30(0.72, 2.34) | 80.8% |
| No | 0 | - | - |

HR, hazard ratio; CI, confidence interval; BMI, body mass index.

affected by multiple factors (age, gender, diabetes, CRP, CV, HbA1c, etc.), but did not explore the relationship between SUA and mortality [40]. In addition, Anderson [41] reported a systematic review and meta-analysis of death risk prediction for patients starting dialysis, unfortunately, the relationship between SUA and death was not explored. A recent meta-analysis by Xue et al. [42] explored the relationship between SUA and all-cause and CVD mortality in PD patients and found that the results of prospective and retrospective cohort studies were inconsistent. However, they did not research the impact of intermediate levels of SUA on mortality compared with the highest and lowest levels. Another meta-analysis by Liu et al. [43] found that high SUA levels were associated with an increased risk of all-cause mortality in PD patients compared with middle SUA levels, but SUA levels may not be associated with CVD mortality, which is consistent with our results before recalculating HRs and 95% CIs. What is more, it may be a pity that both them did not recalculate the original data using the lowest dose group as the reference and perform the dose-response analysis. Liu et al. [7] explored the relationship between patient characteristics and risk factors of early and late mortality in PD patients, which has demonstrated that higher UA level was associated with early death, therefore specific intervention according to risk factors at the initiation of PD should be established to improve the survival of PD patients.

Although the underlying mechanism between SUA and mortality in PD patients is still unclear, some research advances have provided us with clues. Some studies support the

association between high UA levels and high mortality. Animal experiments and clinical studies have confirmed that UA is an endothelial toxin and causes endothelial dysfunction. Hyperuricemia suppresses the production of nitric oxide [44], leading to activation of the renin-angiotensin system, which ultimately leads to endothelial damage [45, 46]. UA-lowing drug (allopurinol and xanthine oxidase inhibitor) treatment resulted in a decrease in SUA. Studies have demonstrated that allopurinol can significantly improve endothelial function in patients with CKD or chronic heart failure [47] and xanthine oxidase inhibitor reduce the incidence of adverse CV events [48]. C-reactive protein (CRP) is the most commonly used inflammatory parameter primarily produced by hepatocytes [49], and elevated CRP levels are independent risk factor for CKD [50]. Evidence suggests that there is a positive correlation between SUA and serum CRP levels in healthy populations, patients with acute coronary syndrome [51], and CKD patients undergoing peritoneal dialysis [52]. CRP is becoming a clinical marker for many noncommunicable diseases (atherosclerosis, CVD, ischemic stroke, hypertension, insulin resistance, and metabolic syndrome) and can independently predict adverse cardiovascular events in individuals, including ischemic stroke, myocardial infarction and sudden cardiac death [53]. In addition, CRP can independently predict all-cause mortality in China's middle-aged and elderly population [54]. The effect of SUA on RRF is an important factor affecting mortality. Elevated SUA is common in patients with PD, and it is inversely related to the decrease in RRF [55]. A study from Taiwan revealed that the UA level has a U-shaped relationship with the decline rate of RRF in continuous ambulatory peritoneal dialysis patients, with a faster decline rate in those of higher and lower UA groups [56]. The rate of decline of RRF is a powerful predictive factor associated with lower survival and technical failure in patients treated with PD [57, 58], which may explain one of the reasons for the effect of both higher and lower SUA concentrations on the death of patients receiving PD treatment.

In addition, research has also reported the role of low UA levels in mortality. Malnutrition is an independent risk factor for the prognosis of patients with PD [23, 59]. Patients with malnutrition have low immunity, are prone to various infections and are difficult to control. In addition, malnutrition is closely related to cardiovascular events [60]. SUA levels are associated with nutritional risk and independently predict all-cause and CVD death risk [39]. Bae et al. [29] have found that SUA < 5.5mg/dL is associated with all-cause mortality. SUA is not only a simple biomarker which indicates the nutritional status of patients treated with chronic dialysis, but also one of the most important antioxidants in human biological fluids, removing excess oxygen free radicals from the body [29]. SUA levels were correlated with the total antioxidant capacity in population treated with dialysis, hence, hypouricemia may lead to a decrease in total antioxidant capacity in dialysis patients [61]. Lai et al. [17] reported that high UA levels were associated with low all-cause and cardiovascular mortality in female populations undergoing continuous PD, which may be explained by the antioxidant capacity of UA. UA plays a major antioxidant role in the plasma, but a major pro-oxidant role when it enters cells and a pro-oxidant role in the development of cardiovascular disease [62].

In summary, research evidence shows that SUA has a dual biological effect on the human body, which may partly explain why the results of the included studies are contradictory and the pooled results are not clear. Furthermore, we need to clarify which effect is greater in PD population. If the harmful effect is greater, we can reduce the concentration of SUA through drugs, diet or other measures. Meanwhile, the protective role of SUA, such as antioxidant capacity, can be played by dietary supplements [62]. In order to clarify the association between SUA and mortality (all-cause and CVD) and to guide clinical treatment, strict design, large sample size and multi-center cohort studies are required to collect as much information as

possible, such as SUA at different time points, changes in SUA level during PD treatment and populations with different characteristics (gender, diabetes and non-diabetes, etc).

Several limitations should be noted. First, we found significant heterogeneity in our study. The "leave-one-out" sensitive analysis indicated that the study conducted by Lai et al. [17] had a great influence on the combined HRs and maybe was the pivotal contributor to heterogeneity, which may be attributed to its opposite results with other studies (a higher SUA level was associated with a lower risk of all-cause and CVD mortality). The reason may be the longer median follow-up period (> 3 years), the relationship between low SUA levels and malnutrition, and increased oxidative stress. Furthermore, results of subgroup analyses suggested that study design (prospective or retrospective, single center or multi-center cohort study), year of publication, sample size, duration of follow-up, male proportion and whether adjusted for diabetes status and BMI may be sources of heterogeneity. It is interesting that there is no significant difference in the subgroup analysis when adjusted with diabetes and BMI, suggesting there is a difference in the impact of UA in these subgroups. Dong et al. [18] found that the associations of UA and CVD/all-cause mortality disappeared with additional adjustment for traditional CV factors such as CVD history, diabetes, BMI, and low-density lipoprotein cholesterol. This may indicate that the association between UA and CVD/all-cause mortality is not independent, but related to traditional CV risk factors in the PD population. At the same time, we found that the pooled HR of SUA and all-cause mortality was significant when the proportion of men was $\geq$ 50%, which was consistent with some previous studies [14, 15, 19]. In addition, UA measured at different time points after PD initiation (including three months [14], sixth month [20] or time-average SUA [19]) may also affect the results. Second, covariate adjustment may affect the correlation between SUA and all-cause and CVD mortality, although we extracted the HRs that adjusted the greatest degree of potential confounders. Furthermore, although most of the original studies adjusted for many important confounding factors, the effects of residual and unknown confounding factors on the results cannot be completely excluded. Finally, all the studies included in the meta-analysis were conducted in Asian countries except one study, which greatly limits the applicability of the results to the global population.

Our analysis has several strengths. First, we comprehensively considered the relationship between different SUA levels (the highest versus the lowest; the highest or lowest versus the median) and mortality. Second, considering the characteristics of SUA, we hypothesized that very low and very high SUA levels may increase the risk of death. Therefore, we recalculated the relevant HRs and 95% CIs using the lowest dose group as the reference, and conducted a dose-response curve instead of only comparing the effects of the highest versus the lowest levels of SUA on mortality to explore the range of SUA concentration associated with the lowest mortality in patients with PD. The dose-response analysis suggested a J-shaped nonlinear relationship between SUA concentrations and all-cause mortality although there was no significant difference in all-cause mortality between the highest and lowest subgroup of SUA level in the main analysis results. This may suggest that there is a group of people with the lowest risk in the distribution of death risk of PD population caused by exposure, so finding them has very important public health significance.

## Conclusion

This meta-analysis did not find there is any relationship between SUA levels and the risk of all-cause and CVD death in PD patients. More rigorously designed studies in the future will be needed to determine the relationship between SUA and cardiovascular and all-cause mortality.

## Supporting information

**S1 Checklist. PRISMA checklist.**
(DOCX)

**S1 Fig. Sensitivity analysis.** For relationship between SUA by categories (the highest SUA category vs the lowest) and all-cause mortality in PD patients before recalculated the HRs and 95% CIs. HR, hazard ratio; CI, confidence interval.
(TIF)

**S2 Fig. Sensitivity analysis.** For relationship between SUA by categories (the highest SUA category vs the median) and all-cause mortality in PD patients before recalculated the HRs and 95% CIs. HR, hazard ratio; CI, confidence interval.
(TIF)

**S3 Fig. Sensitivity analysis.** For relationship between SUA by categories (the lowest SUA category vs the median) and all-cause mortality in PD patients before recalculated the HRs and 95% CIs. HR, hazard ratio; CI, confidence interval.
(TIF)

**S4 Fig. Sensitivity analysis.** For relationship between SUA by categories (the highest SUA category vs the lowest) and cardiovascular mortality in PD patients before recalculated the HRs and 95% CIs. HR, hazard ratio; CI, confidence interval.
(TIF)

**S5 Fig. Sensitivity analysis.** For relationship between SUA by categories (the highest SUA category vs the lowest) and all-cause mortality in PD patients after recalculated the HRs and 95% CIs. HR, hazard ratio; CI, confidence interval.
(TIF)

**S6 Fig. Sensitivity analysis.** For relationship between SUA by categories (the highest SUA category vs the lowest) and cardiovascular mortality in PD patients after recalculated the HRs and 95% CIs. HR, hazard ratio; CI, confidence interval.
(TIF)

**S1 Table. Search strategies for electronic databases.**
(DOCX)

**S2 Table. Quality assessment of the included studies utilizing the Newcastle-Ottawa Scale (NOS).** RCS, retrospective cohort study; PCS, prospective cohort study.
(DOCX)

## Acknowledgments

Thanks to Stanislav Seydou Traore for revising the grammar of this article.

## Author Contributions

**Conceptualization:** Youchun Hu.

**Data curation:** Ting Kang, Youchun Hu, Xuemin Huang.

**Formal analysis:** Ting Kang, Youchun Hu.

**Methodology:** Ting Kang, Youchun Hu.

**Software:** Ting Kang, Youchun Hu.

**Validation:** Adwoa N. Amoah.

**Visualization:** Ting Kang, Youchun Hu.

**Writing – original draft:** Ting Kang.

**Writing – review & editing:** Ting Kang, Youchun Hu, Adwoa N. Amoah, Quanjun Lyu.

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
