## [Decision Letter · Decision Letter 0]

5 Nov 2021

PONE-D-21-22383Serum uric acid level and all-cause and CVD mortality in peritoneal dialysis patients: A systematic review and meta-analysis of cohort studiesPLOS ONE

Dear Dr. Lyu,

Thank you for submitting your manuscript to PLOS ONE. After careful consideration, we feel that it has merit but does not fully meet PLOS ONE’s publication criteria as it currently stands. Therefore, we invite you to submit a revised version of the manuscript that addresses the points raised during the review process.

ACADEMIC EDITOR:High heterogeniety is observed in result section for some outcomes, need for some more statistical analysis as given. Some minor language polishing is required in abstract and the main manuscript

We look forward to receiving your revised manuscript.

Kind regards,

Girish Chandra Bhatt, MD, FASN

Academic Editor

PLOS ONE

Journal Requirements:

Additional Editor Comments:

Details of how NOS quality assessment of each study should be provided in the table.

Wherever unexplained high heterogeneity is present i.e >50% , the authors should use methods such as Bejout;s curve, influential analysis and leave one out analysis.

Visual inspection of the funnel plot for publication bias is used when the number of the studies is more than or equal to 10 for a particular outcome. Moreover, other test such as Egger;s test should be usede and provided as being significant or non -significant.

Reviewers' comments:

Reviewer's Responses to Questions

**Comments to the Author**

1. Is the manuscript technically sound, and do the data support the conclusions?

Reviewer #1: Yes

2. Has the statistical analysis been performed appropriately and rigorously? 

Reviewer #1: Yes

3. Have the authors made all data underlying the findings in their manuscript fully available?

Reviewer #1: Yes

4. Is the manuscript presented in an intelligible fashion and written in standard English?

Reviewer #1: Yes

5. Review Comments to the Author

Reviewer #1: Serum uric acid level and all cause mortality and cardiovascular mortality in peritoneal dialysis patients: A systematic review and meta-analysis of cohort studies.

Introduction:

The introduction highlights the facts that serum uric acid is a know cause on increased mortality in patients on dialysis. There is conflicting data on the association of uric acid with mortality in patients on PD and the method of estimation of uric acid. The authors may re-frame the sentences to allow easy comprehension and improve clarity.

The objective may also be re-framed “to determine the association…..”

Methods:

The methods mention that studies in all languages were included, but the interpretation of studies in a foreign language has not been described. Were any attempts made to search for unpublished data (conference proceedings etc?). Cochrane library was not included in the literature search. The systematic review /meta-analysis has not been registered in any database.

The systematic review has assessed the bias of risk, assessed quality of the data and accounted for publication bias.

Results:

The results have been discussed well. The Forest plots however are not very clear.

The PRISMA reporting guidelines have been followed.

Discussion: The fact that most of the studies were from Asian countries limits its generalizability to the global population

Conclusions: The conclusions are aligned to the objectives

6. PLOS authors have the option to publish the peer review history of their article (what does this mean?). If published, this will include your full peer review and any attached files.

Reviewer #1: No

---

## [Author Response · Author response to Decision Letter 0]

9 Dec 2021

Response to Academic Editor:

To explore the source of heterogeneity among studies, subgroup analyses were conducted according to study design (prospective or retrospective cohort study), number of center (multi-center or single center), publication years (2013-2016 or 2017-2020), sample size (< 900 or > 900), follow-up duration (< 30 months or > 30 months), adjustment for gender and adjustment for diabetes status. And we also carefully revised the content and grammar of our article, hoping to improve the article.

Response to additional Editor:

To question 1: We provided details of NOS quality assessment for each study in the S2 Table.

To question 2: Sensitivity analysis, in which 1 study was removed at a time, was performed to evaluate the stability of the results. We added several sensitivity analyses, and the results are shown in S1-S6 Figs.

To question 3: It is really true as editor suggested that visual inspection of the funnel plot for publication bias is used when the number of the studies is more than or equal to 10 for a particular outcome. Therefore, we deleted the funnel plots and recorded the P values of egger’s test in the manuscript.

Response to Reviewer:

To question 1: We re-framed the objective of this study as follows: hence, the objective of this study is to determine the association between SUA and all-cause and CVD mortality with a detailed analysis of eligible literature. 

To question 2: 

(1) We have re-written this part according to the Reviewer’s suggestion: all non-English studies were translated by software first, and then the researchers checked whether they could be included.

(2) In addition to a comprehensive search, we considered the reference lists of all eligible articles in order to find additional relevant articles. I regret that unpublished data was not considered.

(3) We are very sorry that we did not search the Cochrane Library database before. Therefore, we limited the date filter to April 7, 2021, and finally retrieved 27 literatures (See S1 Table for searching strategies).

(4) We are very sorry that we did not register for this meta-analysis, but we conducted this systematic review and meta-analysis in strict accordance with the Preferred Reporting Items for Systematic Reviews and Meta-Analyses (PRISMA) statement.

To question 3: We have adjusted the resolution and size of the figures, hoping to meet the requirements of the journal.

To question 4: After consideration, we have included a conference abstract by Coelho et al [1] in Portugal, but the pooled results did not change. Nevertheless, almost all the studies included in the meta-analysis were conducted in Asian countries, which greatly limits the applicability of the results to the global population.

---

## [Decision Letter · Decision Letter 1]

21 Jan 2022

PONE-D-21-22383R1Serum uric acid level and all-cause and cardiovascular mortality in peritoneal dialysis patients: a systematic review and dose-response meta-analysis of cohort studiesPLOS ONE

Dear Dr. Lyu,

Thank you for submitting your manuscript to PLOS ONE. After careful consideration, we feel that it has merit but does not fully meet PLOS ONE’s publication criteria as it currently stands. Therefore, we invite you to submit a revised version of the manuscript that addresses the points raised during the review process. The manuscript is still necessary to be revised to be improve the paper quality according to the Reviewer's comments.

We look forward to receiving your revised manuscript.

Kind regards,

Masaki Mogi

Academic Editor

PLOS ONE

Reviewers' comments:

Reviewer's Responses to Questions

**Comments to the Author**

1. If the authors have adequately addressed your comments raised in a previous round of review and you feel that this manuscript is now acceptable for publication, you may indicate that here to bypass the “Comments to the Author” section, enter your conflict of interest statement in the “Confidential to Editor” section, and submit your "Accept" recommendation.

Reviewer #1: All comments have been addressed

Reviewer #2: (No Response)

2. Is the manuscript technically sound, and do the data support the conclusions?

Reviewer #1: Yes

Reviewer #2: Yes

3. Has the statistical analysis been performed appropriately and rigorously? 

Reviewer #1: Yes

Reviewer #2: Yes

4. Have the authors made all data underlying the findings in their manuscript fully available?

Reviewer #1: Yes

Reviewer #2: Yes

5. Is the manuscript presented in an intelligible fashion and written in standard English?

Reviewer #1: Yes

Reviewer #2: Yes

6. Review Comments to the Author

Reviewer #1: The authors have responded to all the queries satisfactorily and made the necessary changes in the revised manuscript

Reviewer #2: The authors conducted a very careful systematic review and presented valuable data on association between serum uric acid levels and all-cause mortality or CVD in PD patients. It is very interesting data, and since this study contains enough subjects, subgroup analysis became possible. So, several concerns should be addressed.

Major comments

#1 It is interesting that there is no significant difference in the subgroup analysis (Table 1 and Table2) when adjusted with gender and diabetes, suggesting there is a difference in the impact of uric acid in these subgroups. Are such analyses possible? Or, authors should discuss on these points and data of subgroup analyses (Table 1, Table2).

#2 The patients with hyperuricemia consists of etiologically heterogeneous population. Serum uric acid levels are affected by genetic predisposition for uric acid transporters, and also by environmental factors such as alcohol drinking and obesity/visceral fat accumulation due to overnutrition. The former shows a very high serum uric acid level, and gout and urinary tract stones appear in the foreground, while the latter is prone to atherosclerosis, and may reflect the high CRP that the authors pointed out in Discussion section. Since this study have enough subjects, subgroup analysis is possible, which is very valuable. This time, authors showed the adjusted data by gender and diabetes. Since the authors focused on all-cause mortality and CVD, it may be useful to conduct a subgroup analysis with or without obesity/visceral fat accumulation, and with or without liver dysfunction which reflects NAFLD/NASH. Is it possible that the impact of uric acid will be sharpened by doing so?

Minor comments

#3 It will become more understandable for readers to show data on the concentration range of the uric acid category in each paper, because the meanings of UA<2 and UA <4 are different in terms of dual biological effects of uric acid. Is there a difference in the width of the lowest between each paper?

#4 In that sense, the dose-response analysis data is very significant. Is it possible to show the data related to CVD as a Figure (even if it is negative data)? Author had better to discuss more this dose-response impact on all-cause mortality and CVD in discussion section.

#5 Figure 2 is hard to see. So, authors should increase the resolution.

7. PLOS authors have the option to publish the peer review history of their article (what does this mean?). If published, this will include your full peer review and any attached files.

Reviewer #1: No

Reviewer #2: No

---

## [Author Response · Author response to Decision Letter 1]

6 Feb 2022

Response to Reviewer 2:

Question 1: It is interesting that there is no significant difference in the subgroup analysis (Table 1 and Table2) when adjusted with gender and diabetes, suggesting there is a difference in the impact of uric acid in these subgroups. Are such analyses possible? Or, authors should discuss on these points and data of subgroup analyses (Table 1, Table2)

Answer: First, we carefully checked the variable adjustment of each included article, and ran the heterogeneity test again with Stata software, and the results were consistent with those in the manuscript (Table 2 and Table 3). Dong et al. [1] found that the associations of UA and CVD/all-cause mortality disappeared with additional adjustment for traditional CV factors such as CVD history, diabetes, BMI, and low-density lipoprotein cholesterol. This may indicate that the association between UA and CVD / all-cause mortality is not independent, but related to traditional CV risk factors in the PD population. Second, after careful consideration, we changed the subgroup analysis of whether to adjust gender into the subgroup analysis of male proportion (50% as the cut-off value). As written in paragraph 4 of the “result” section, there may be gender differences in the impact of SUA levels on mortality. Therefore, subgroup analysis based on the proportion of men may be more reasonable. We discussed the results of the subgroup analysis in lines 460-469.

Question 2: Since the authors focused on all-cause mortality and CVD, it may be useful to conduct a subgroup analysis with or without obesity/visceral fat accumulation, and with or without liver dysfunction which reflects NAFLD/NASH. Is it possible that the impact of uric acid will be sharpened by doing so?

Answer: Thank you for your suggestion. We conducted a subgroup analysis according to adjustment for BMI, but the results were similar to those adjusted for diabetes. After careful review of the included literature, NAFLD / NASH was not found in the comorbid conditions of the collected information. For defining NAFLD [2], there must be (1) evidence of hepatic steatosis, either by imaging or histology, and (2) lack of secondary causes of hepatic fat accumulation such as significant alcohol consumption, long-term use of a steatogenic medication, or monogenic hereditary disorders. Unfortunately, there is no data of liver imaging and histological examination in the original article. A recent meta-analysis [3] indicated that controlling the most important confounding factors, including indicators that reflect the current residual renal function of patients, and other confounding factors (including gender, age, diabetes history, CVD history, Kt/V, whether to use UA drugs and serum albumin) are very important for comparability among groups. We noticed that albumin is an index that can reflect the synthesis and reserve function of the liver in a certain period of time. Recent studies have shown NAFLD to be associated with impairments in albumin function, which are associated with impairments in liver function and disease prognosis [4]. However, we found that all studies included in this meta-analysis had adjusted for albumin, so subgroup analysis was not performed. 

Question 3: It will become more understandable for readers to show data on the concentration range of the uric acid category in each paper.

Answer: Thank you very much for your advice. We have added concentration range of the SUA categories in Table 1. 

Question 4: (1) Is it possible to show the data related to CVD as a Figure (even if it is negative data)? (2) Author had better to discuss more this dose-response impact on all-cause mortality and CVD in discussion section.

Answer: (1) We examined the dose-response relationship between SUA and CVD mortality and confirmed that the dose-response diagram could not be made. The P values of model goodness-of-fit tests were all < 0.05, suggesting that heterogeneity had to be considered. But neither liner (�2 = 0.84, P = 0.360, random effect linear model) nor nonlinear (�2 = 1.66, P = 0.476, random effect nonlinear model) relationship between SUA and CVD mortality was observed. The reason why there is no dose-response figure of SUA and CVD is that the random effect model is not significant, rather than the curve. Therefore, we deleted the sentence " We analyzed the linear or non-linear relationship between mortality and SUA levels, and listed only significant curves." in the “statistical analysis” part because it is very misleading. 

(2) Thank you very much for your advice. We added the discussion about the dose-response relationship between SUA levels and all-cause mortality. Please see lines 483-492 for details.

Question 5: Figure 2 is hard to see. So, authors should increase the resolution.

Answer: We have adjusted the resolution of figure2, hoping to meet the requirements of the journal.

Once again, thank you very much for your comments and suggestions. Looking forward to hearing from you. Best wishes.

---

## [Decision Letter · Decision Letter 2]

9 Feb 2022

Serum uric acid level and all-cause and cardiovascular mortality in peritoneal dialysis patients: a systematic review and dose-response meta-analysis of cohort studies

PONE-D-21-22383R2

Dear Dr. Lyu,

We’re pleased to inform you that your manuscript has been judged scientifically suitable for publication and will be formally accepted for publication once it meets all outstanding technical requirements.

Kind regards,

Masaki Mogi

Academic Editor

PLOS ONE

Additional Editor Comments (optional):

No further comment.

Reviewers' comments:

Reviewer's Responses to Questions

**Comments to the Author**

1. If the authors have adequately addressed your comments raised in a previous round of review and you feel that this manuscript is now acceptable for publication, you may indicate that here to bypass the “Comments to the Author” section, enter your conflict of interest statement in the “Confidential to Editor” section, and submit your "Accept" recommendation.

Reviewer #2: All comments have been addressed

2. Is the manuscript technically sound, and do the data support the conclusions?

Reviewer #2: Yes

3. Has the statistical analysis been performed appropriately and rigorously? 

Reviewer #2: Yes

4. Have the authors made all data underlying the findings in their manuscript fully available?

Reviewer #2: Yes

5. Is the manuscript presented in an intelligible fashion and written in standard English?

Reviewer #2: (No Response)

6. Review Comments to the Author

Reviewer #2: All comments, which the reviewer had raised, have adequately addressed.

The reviewer had no more question and comments.

7. PLOS authors have the option to publish the peer review history of their article (what does this mean?). If published, this will include your full peer review and any attached files.

Reviewer #2: No

---

## [Editor Report · Acceptance letter]

11 Feb 2022

PONE-D-21-22383R2 

Serum uric acid level and all-cause and cardiovascular mortality in peritoneal dialysis patients: a systematic review and dose-response meta-analysis of cohort studies 

Dear Dr. Lyu:

I'm pleased to inform you that your manuscript has been deemed suitable for publication in PLOS ONE. Congratulations! Your manuscript is now with our production department. 

Kind regards, 

on behalf of

Dr. Masaki Mogi 

Academic Editor

PLOS ONE